# Safety and DIVA Capability of Novel Live Attenuated Classical Swine Fever Marker Vaccine Candidates in Pregnant Sows

**DOI:** 10.3390/v16071043

**Published:** 2024-06-28

**Authors:** Chao Tong, Alice Mundt, Alexandra Meindl-Boehmer, Verena Haist, Andreas Gallei, Ning Chen

**Affiliations:** 1Boehringer Ingelheim Vetmedica (China) Co., Ltd., No. 299, Xiangtai Road, Taizhou 225300, China; chao.tong@boehringer-ingelheim.com; 2Boehringer Ingelheim Vetmedica GmbH, Binger Str. 173, 55216 Ingelheim am Rhein, Germany; alice.mundt@boehringer-ingelheim.com (A.M.); alexandra.meindl-boehmer@boehringer-ingelheim.com (A.M.-B.); verena.haist@boehringer-ingelheim.com (V.H.); andreas.gallei@boehringer-ingelheim.com (A.G.)

**Keywords:** classical swine fever, DIVA vaccine, pregnant sow, safety, vertical transmission

## Abstract

Classical Swine Fever (CSF), a highly contagious viral disease affecting pigs and wild boar, results in significant economic losses in the swine industry. In endemic regions, prophylactic vaccination and stamping-out strategies are used to control CSF outbreaks. However, sporadic outbreaks and persistent infections continue to be reported. Although the conventional attenuated CSF vaccines protect pigs against the disease, they do not allow for the differentiation of infected from vaccinated animals (DIVA), limiting their use as an eradication tool. In this study, three targeted attenuation strategies were employed to generate vaccine candidates based on the current prevalent CSFV group 2 strains GD18 and QZ07: a single deletion of H79 in E^rns^ (QZ07-sdErnsH-KARD), double deletion of H79 and C171 in E^rns^ (GD18-ddErnsHC-KARD and QZ07-ddErnsHC-KARD), and deletion of H79 in E^rns^ combined with a 5–168 amino acids deletion of N^pro^ (GD18-ddNpro-ErnsH-KARD). Additionally, a negative serological marker with four substitutions in a highly conserved epitope in E2 recognized by the monoclonal antibody 6B8 was introduced in each candidate for DIVA purposes. The safety of these four resulting vaccine candidates was evaluated in pregnant sows. Two candidates, GD18-ddErnsHC-KARD and QZ07-sdErnsH-KARD were found to be safe for pregnant sows and unlikely to cause vertical transmission. Both candidates also demonstrated potential to be used as DIVA vaccines, as was shown using a proprietary blocking ELISA based on the 6B8 monoclonal antibody. These results, together with our previous work, constitute a proof-of-concept for the rational design of CSF antigenically marked modified live virus vaccine candidates.

## 1. Introduction

Classical swine fever (CSF) is a highly contagious disease affecting pigs and wild boar, leading to significant economic losses [1]. This disease is caused by the classical swine fever virus (CSFV), which belongs to the genus Pestivirus in the Flaviviridae family [2]. Although the World Organization for Animal Health (WOAH) has approved CSF-free territories including 38 countries, covering all North America and Oceania, as well as a large part of the European Union (EU), recent outbreaks were documented in Brazil, Colombia, Russia, India, Korea, and Japan [3,4]. In China, genotype 2.1 strains are predominantly prevalent and responsible for sporadic outbreaks [5,6,7,8,9]. A combination of prophylactic vaccination and stamping-out strategies is employed to control CSF outbreaks in endemic areas [3,6,10,11]. However, despite these measures, sporadic outbreaks and persistent infections continue to be reported [6,8,9,12,13,14]. The widely used conventional CSF modified live vaccines (MLV) effectively protect pigs against CSF, providing early and long-lasting immune responses, full protection, and are safe to pregnant sows [6,15,16]. However, a major limitation of this kind of vaccine is its inability to allow for the differentiation of infected from vaccinated animals (DIVA) using serological tests [7,17,18], hindering its use as an efficient eradication tool. Therefore, there is an urgent need for the development of novel CSF vaccines that maintain the good efficacy profile of conventional ones while offering DIVA capability after vaccination [7,19].

In recent years, numerous approaches have been employed to develop the novel CSF marker MLV [7,19]. An EU-licensed CSF marker vaccine, Suvaxyn^®^ CSF Marker, was developed using a BVDV backbone with a substituted N-terminal E2 [20]. It has an avirulent mechanism in pigs that remains unclear [19,20]. The safety and DIVA features of the LOM-based chimeric vaccine, recently authorized for market use in South Korea, are still under investigation in the field [21,22]. Several vaccine candidates have been generated using a C-strain vaccine seed as a backbone, including C-DIVA and rHCLV-E2P122A [18,23,24].

Targeted attenuation is one strategy in developing safe CSF MLV candidates [7]. Numerous virulence determinants common in members of the *Pestivirus genus*, including nonstructural protein N^pro^ and structural protein E^rns^, have been thoroughly investigated [1]. The N^pro^ protein was shown to counteract the IFN-I/III responses and inhibit the host’s immune response to the virus, allowing the establishment of productive infection [25,26]. The E^rns^ protein is an endoribonuclease (RNase) that degrades RNA molecules. This RNase activity is thought to play a crucial role in the life cycle of the virus, particularly in its evasion of the host’s immune response [27,28]. Attenuation of pestiviruses, including CSFV, has been observed after the deletion of either histidine 79 or cysteine 171 in E^rns^ [29,30,31]. Similarly, the attenuation of a pestivirus was also achieved by deleting the nonstructural protein N^pro^ [32]. Based on these insights, a commercial bovine viral diarrhea (BVD) vaccine was successfully developed, by deleting N^pro^ and histidine 30 in E^rns^ [33]. Furthermore, our lab has reported a genotype 2.1 CSF marker MLV candidate, attenuated by deleting two virulence-associated functional residues in E^rns^, namely histidine 79 and cysteine 171 [34]. This candidate demonstrated its efficacy by protecting pigs against lethal challenge with the highly virulent Shimen strain. An overdose safety study confirmed its safety in piglets. Importantly, the infected animals can be differentiated using the accompanying DIVA ELISA [34].

According to Chapter 3.9.3 “Classical swine fever” of the Terrestrial Manual of the WOAH [35], a fully attenuated CSF vaccine candidate must be shown to be safe in pregnant sows and show no vertical transmission [1,21,36,37]. Vertical transmission from sow to fetuses occurs frequently after CSFV infection and may cause safety issues, such as an early infection in the uterus, abortion, and mummification [36,38]. Vertical transmission can also cause persistent infection in piglets. Those piglets shed virus for extended periods of time while they do not respond to CSF vaccination [21,36,39]. Therefore, ensuring the safety of a vaccine in pregnant sows is of utmost importance when developing a next-generation CSF marker MLV.

The current study builds upon a previously reported CSF marker MLV that showed promising safety, efficacy, and DIVA potential in piglets [34]. Four CSF marker MLV candidates were generated, including the one previously mentioned. For these candidates, the Chinese prevalent 2.1 genotype strains GD18 and QZ07 were used as backbones for three different targeted attenuation strategies [5,6,40]: the double deletion of H79 and C171 in E^rns^ as previously tested [34], single deletion of E^rns^ H79 which was shown to have abolished RNase activity, and deletion of E^rns^ H79 combined with deletion of amino acids 5–168 of N^pro^. Additionally, a negative serological marker consisting of four amino acid substitutions (S14K, G22A, E24R, and G25D) in a highly conserved epitope in the B/C antigenic domain of E2 was introduced into each construct for DIVA purposes [34,40]. Safety and DIVA features in pregnant sows of these four MLV candidates were investigated in this study.

## 2. Materials and Methods

### 2.1. Cell Lines and Virus

The AI-ST A1 and PK/WRL cell lines are available at Boehringer Ingelheim as previously reported [6,34]. The PK/WRL cell line was grown in Minimal Essential Media (MEM) (Sigma-Aldrich, St. Louis, MO, USA) supplemented with 10% Fetal Bovine Serum (FBS) (Sigma-Aldrich), free of pestiviruses and their antibodies. The AI-ST A1 cell line was cultivated using the same MEM supplemented with 5% FBS. Both cell lines were cultured at 37 °C and 5% CO_2_ in a humidified incubator. The moderately virulent CSFV isolate GD18 was obtained from the Institute of Military Veterinary Medicine, Changchun, China, whereas the low virulent strain QZ07 was purchased from Zhejiang University, Hangzhou, China [5,6,40]. The genetically modified CSFV Alfort-Erns H297K (H297 of the polyprotein of CSFV equals H30 in E^rns^) was kindly provided by Gregor Meyers (Friedrich-Loeffler-Institute, Greifswald-Insel Riems, Germany). This virus was reported as partially attenuated by a histidine to lysine mutation at position 30 in the E^rns^ protein; however, it still causes morbidity in piglets [31].

### 2.2. Plasmids and Antibodies

The plasmids pBeloBAC, containing the approximate 3′ half of the HDV ribozyme (36 bp downstream of the *Rsr*II restriction site, cggtccgacctgggctacttcggtaggctaagggag), and pACYC are available at Boehringer Ingelheim, as previously published [34]. The CSFV E2-targeting 6B8 monoclonal antibody (mAb) recognizes a conserved conformational epitope in the B/C domain of the CSFV E2 glycoprotein and recognizes both genogroups 1 and 2 CSFV strains, as described previously [34,40]. The anti-CSFV rabbit polyclonal antibody (poly Ab) is Boehringer Ingelheim property [34].

### 2.3. Construction of Infectious Clones

As previously described [34], the full-length cDNA infectious clones of the GD18 and QZ07 strains were subcloned into three overlapping fragments: a 2.2-kb 5′-end fragment, a 10-kb middle segment, and a 2-kb 3′-end fragment. A T7 promoter sequence was placed upstream of the 5′UTR in the first fragment, whereas the HDV ribozyme and a *Pac*I restriction enzyme site were inserted downstream of the 3′UTR for in vitro transcription. Based on the QZ07 strain, two types of deletion constructs were generated. In the 2.2-kb 5′-end fragment either a single deletion of E^rns^ H79 or a double deletion of E^rns^ H79 and C171 were introduced. On the basis of the GD18 strain, two types of deletion constructs were generated. In the 2.2-kb 5′-end fragment either a double deletion of N^pro^ (amino acid residues 5–168) and E^rns^ H79 or a double deletion of E^rns^ H79 and C171 were introduced. To remove the 6B8 conformational epitope from the E2 protein of each strain, four substitutions (S14K, G22A, E24R, and G25D) were introduced in the full-length infectious clone plasmids using a two-step Red-mediated recombination method to generate the final constructs [41]. In total, four infectious clones were generated, named as follows: GD18-ddNpro-ErnsH-KARD, GD18-ddErnsHC-KARD, QZ07-ddErnsHC-KARD, and QZ07-sdErnsH-KARD (where “dd” denotes double deletion and “sd” denotes single deletion). Detailed information on these constructs can be found in Table 1. 

### 2.4. Rescue and In Vitro Characterization of Vaccine Candidates

A minimum of 1 µg of linearized plasmid DNA of each of the four infectious clones (GD18-ddNpro-ErnsH-KARD, GD18-ddErnsHC-KARD, QZ07-ddErnsHC-KARD, and QZ07-sdErnsH-KARD) was used as a template for in vitro transcription, following a previously described protocol [34]. Each virus was rescued by electroporating 5 μg of RNA transcript into 2 × 10^6^ PK/WRL cells at 150 V and 500 μF. After electroporation, the cells were incubated at 37 °C and 5% CO_2_ for 4 h, after which the medium was replaced to remove dead cells. After five days of incubation, cells were frozen and thawed three times. The cell suspension was centrifuged at 8000 rpm for 10 min and supernatant was collected. This supernatant was designated as the first passage (P1) of the rescued virus. Each recombinant virus was passed by infecting PK/WRL cells at 50–60% confluency with a multiplicity of infection (MOI) of 0.01. Infected cells were incubated at 37 °C and 5% CO_2_ for five days. Different passages of each virus were stained with the 6B8 mAb and poly Ab for characterization as described previously [34]. The replication kinetics of each rescued virus were investigated using passage 6 (P6), peak titers were determined as described previously [34] and each virus stock was only tested one time for kinetics. The recombinant viruses were serially passed as described until passage 10 (P10). Viral genomic RNA from P2 and P10 was reverse transcribed, and the presence of the introduced mutations was confirmed by PCR and sequencing, as described previously [34].

To generate the recombinant viruses to be tested as vaccine candidates in the animal study at Boehringer Ingelheim Veterinary Research Center (Hannover, Germany), all four constructs were rescued in AI-ST A1 cells using the same process. This was necessary, as AI-ST A1 cells fulfill the required criteria for a production cell line. 

### 2.5. Animal Study 

The experiment was designed according to the WOAH Terrestrial Manual Chapter 3.9.3 [35], specifying safety testing for CSF vaccines in pregnant sows. According to these guidelines, vaccination should be carried out between the 55th and 70th day of gestation, using a quantity not less than the maximum virus titer likely to be contained in one dose of the final vaccine. The schedule for the pregnant sow study is depicted in Figure 1.

#### 2.5.1. Animal Sourcing and Maintenance

A total of 40 commercial mix (PIC genetics) pregnant sows were purchased. All of these sows were in the second trimester of gestation (55th or 56th day of gestation) at the beginning of the study (D0) and had previously been vaccinated against porcine parvovirus and porcine circovirus 2. Of these, 35 pregnant sows were transported to the animal facility of the Boehringer Ingelheim Veterinary Research Center (Hannover, Germany) at approximately seven weeks of gestation, under controlled conditions. During the acclimatization period, the sows were housed in groups in two separate rooms. Upon randomization, three days before vaccination, the animals were moved to distinct rooms, with each of the treatment groups 1–5 assigned to one of the rooms (Table 2). Water and feed of appropriate quality were available ad libitum and were managed according to the specific requirements of the pigs at their age and gestational status. All serum samples taken from the 35 pregnant sows at three days before inoculation (D-3) were tested for CSFV and bovine viral diarrhea virus (BVDV) antibody and genome by ELISA and reverse transcriptase quantitative PCR (RT-qPCR), respectively, as well as for porcine reproductive and respiratory syndrome virus and swine influenza virus genomes. The remaining five sows of group 6 were kept at the farm of origin and were housed under standard field conditions. 

#### 2.5.2. Experimental Design

Detailed information on the treatment and actions for each group is provided in Table 2. Sows in groups 1–4 were vaccinated intramuscularly at 55th or 56th days of gestation with one of the respective vaccine candidates produced in AI-ST A1 cell line. Sows belonging to group 5 served as a control and received an intramuscular injection of the genetically modified CSFV isolate Alfort-Erns H297K, which is known to cross the placenta and cause reproductive disorders such as the mummification of fetuses. Approximately 42–45 days after vaccination/treatment, all sows of groups 1–5 were euthanized to collect data from the fetuses. 

Study parameters included clinical signs after the administration of the vaccine candidates (such as injection site reactions and rectal temperature), clinical signs related to gestation (e.g., abortion) and the presence of the vaccine virus or antibodies against CSFV in the blood and tissue samples from the fetuses of groups 1–5. The study was to be considered valid if sows seroconverted against CSFV in each tested group. The WOAH Terrestrial Manual Chapter 3.9.3 recommends following animals up to birth and testing newborn piglets for the presence of vaccine virus and antibodies against CSFV before the ingestion of colostrum [35]. In this study, the protocol deliberately deviated from this requirement due to animal welfare reasons. Sows were euthanized between the 98th and 101st day of gestation, and their fetuses were obtained, assessed, sampled, and analyzed at this stage and not after birth. Whole blood samples from sows were collected at D -3, D7, and D14 and at necropsy. They were stored at 4 °C overnight for blood clotting and centrifuged at 10,000× *g* for 10 min for serum collection. Those samples were analyzed by RT-qPCR to detect the genome of the vaccine candidates. Additionally, the presence of CSFV antibodies in the sera was tested by ELISA using serum samples collected at D -3 and at necropsy.

The five sows in group 6 were housed under standard field conditions in the farm of origin until farrowing. Serum sampling of these sows was done within two weeks after farrowing. Those samples were analyzed by RT-qPCR for viremia and antibodies. Additionally, reproductive performance parameters (abortions, stillborn piglets, mummies, the number of live born, and the number of piglets alive at day 5 after farrowing) were collected.

#### 2.5.3. Detection of CSFV Genome in Fetuses 

To evaluate the potential for vertical transmission of each candidate, at the end of the study, samples of tonsils, umbilical cord blood, thymus, spleen, kidney, and small intestine were collected from each fetus in groups 1–5, except in cases where it was not possible due to conditions such as mummification. In these cases, a variety of soft tissues was collected, with a preference for tonsil and/or thymus, if identifiable. CSFV-specific RT-qPCR testing of blood samples and tissue homogenate extracts was performed by the WOAH and EU CSFV Reference Laboratory (University of Veterinary Medicine Hannover, Hannover, Germany) with each sample tested in duplicate. Total cellular RNA was prepared directly from serum or from tissue homogenates. Tissue homogenates were prepared in a BeadMill24 (Thermofisher, Waltham, MA, USA) homogenizer using lysing Matrix M (MP Bio, Eschwege, Germany) in a lysis buffer RAl (Macherey/Nagel, Düren, Germany) containing β-Mercaptoethanol. RNA was prepared on the automated extraction platform Kingfisher DuoPrime (Thermofisher, Waltham, MA, USA) using an IndiMag Pathogen Kit (Indical Biosciences, Leipzig, Germany) according to the manufacturer’s recommendations. CSFV genome detection was performed in a Taqman-probe-based RT-PCR assay using well-established primers targeting the 5′ non-translated region (NTR) of the viral genome as described in [42]. An internal control was added to the lysed homogenates, immediately prior to extraction, to control false negative results due to the possible presence of PCR inhibitors or insufficient RNA extraction. A test was regarded as positive if specific fluorescence was detected in both duplicates within 40 rounds of amplification (Ct < 40). In addition, a test was considered doubtful if the duplicates produced one result showing one Ct value higher than 36 but lower than 40 and the other Ct value was equal to or higher than 40. Positive and doubtful samples were re-tested using either conventional RT-PCR for the amplification of a short 5′ terminal genomic fragment or a 5′ terminal fragment of the E2 gene, or by SYBR green-based RT-qPCR amplifying a 288-nucleotide fragment from the 5′ end of the genome [43].

#### 2.5.4. Serological Tests of Sows and Fetuses

##### CSFV Ab Test for CSFV Antibody Detection 

Serum samples were collected from sows in groups 1–5 three days before vaccination and on the day of necropsy to detect CSFV antibodies. From the fetuses, serum was prepared from umbilical cord blood. From the sows of the negative control group kept on the farm, serum samples were collected within two weeks after farrowing. Sera were tested using a commercial CSFV Ab test (IDEXX Laboratories, Westbrook, ME, USA) according to the manufacturer’s instructions, which included an overnight incubation of the sample material in ELISA plates. All analyses were performed in duplicate and OD_450_ was measured using a Synergy 2 microplate reader (BioTek, Winooski, VT, USA).

##### Double-Competitive DIVA ELISA (6B8 dcELISA) Testing for DIVA Purposes

Serum samples collected on the day of necropsy from sows in groups 1–5 were also analyzed using a proprietary DIVA ELISA as previously described [34]. OD_450_ was measured using a Synergy 2 microplate reader (BioTek, Winooski, VT, USA). The cutoff value of the positive sample was set as >25% blocking rate. Due to the introduction of the KARD mutations in the vaccine candidates, negative results with low blocking rates against the 6B8 mAb were expected. For the control group (group 5), a positive result with a high blocking rate was expected [34].

### 2.6. Statistical Analysis

Data were summarized in tables and/or graphs. The outcome parameters were analyzed using appropriate descriptive graphical and numerical summary statistics, categorized by treatment group.

## 3. Results

### 3.1. All Vaccine Candidate Viruses Were Rescued In Vitro and Were Genetically Stable

Transfected cells were incubated at 37 °C and 5% CO_2_ for five days, followed by indirect immunofluorescence assay (IFA) using poly Ab to determine transfection efficiency. Upon the confirmation of successful transfection, the P1 viruses were passaged in PK/WRL cells for nine continuous passages until P10. Fluorescence microscopic images of cell cultures infected with P2 and P10 viruses and incubated with either the 6B8 mAb or poly Ab are shown in Figure 2. Only the poly Ab specifically reacted with infected cells, no signal was observed in cells infected with vaccine candidate viruses when stained with the 6B8 mAb. These results confirmed the successful rescue of the recombinant virus and the stability of the E2 mutation, which abolished reactivity with the 6B8 mAb. The sequencing results of each P10 virus confirmed the stability of the introduced deletions in E^rns^ or N^pro^ as well as the DIVA marker in E2 protein (Figure 3).

### 3.2. Rescued Candidate Viruses Showed Robust Replication In Vitro 

The P6 stock of each candidate virus was selected for virus growth kinetics based on an MOI of 0.01 in the PK/WRL cell line. Each virus stock was tested once. The purpose of this experiment was to determine the optimal time for harvest to obtain the highest possible titer. As shown in Figure 4, the QZ07-sdErnsH-KARD exhibited a peak titer of 10^7.40^ TCID_50_/mL at 96 h post infection (h.p.i.) and maintained this titer until 144 h.p.i. The GD18-ddNpro-ErnsH-KARD and the QZ07-ddErnsHC-KARD showed similar peak titers of approximately 10^6.0^ TCID_50_/mL; however, the peak titers occurred at different times (96 h.p.i. vs. 120 h.p.i.). The GD18-ddErnsHC-KARD showed the lowest titer of 10^5.1^ TCID_50_/mL, a result consistent with the previous report indicating that P9 of GD18-ddErnsHC-KARD in the AI-ST cell line produced a peak titer of 10^5.1^ TCID_50_/mL at 120 h.p.i. [34]. 

For the present animal study, the vaccine candidates were rescued in the AI-ST A1 cell line, a subclone of AI-ST cells, fulfilling the requirements of a cell line for veterinary vaccine production. Rescued viruses were passaged until their titers were sufficient to be used in the study. QZ07-ddErnsHC-KARD and QZ07-sdErnsH-KARD reached high titers at P2, while GD18-ddNpro-ErnsH-KARD and GD18-ddErnsHC-KARD needed to be passaged to P7 until sufficient titers were obtained.

### 3.3. Pregnant Sow Safety Study Results

#### 3.3.1. Validity of the Study 

The sows tested negative for the presence of CSFV and BVDV genomes and antibodies prior to vaccination, the untreated control group 6 remained negative until the end, and no influential circumstances were detected which might have interfered with the outcome of the study. According to the WOAH Terrestrial Manual Chapter 3.9.3, a study is considered valid if all vaccinated sows of a group seroconvert against CSFV [35]. Seroconversion rates of the groups were calculated to assess the immunogenicity of each vaccine candidate as well. All sows (100%) in groups 1, 2, 4, and 5 were clearly positive for CSFV antibodies in the IDEXX ELISA (Table 3 and Figure 5A). However, only four sows out of seven in group 3, vaccinated with GD18-ddNpro-ErnsH-KARD, had CSFV antibody titers above the threshold, while three were below. One sow in group 3 showed severe lameness on her right front leg with obvious depression. This animal was euthanized on D14 for animal welfare reasons. Overall, the study was considered valid for groups 1, 2, 4, and 5. It was not valid for group 3; therefore, the data of this group were not included in the analysis.

#### 3.3.2. Candidates in Groups 1, 2, and 4 Showed Promising Safety Profiles in Pregnant Sows and Fetuses

None of the vaccinated sows in groups 1, 2, and 4 displayed any injection site reaction after vaccination, nor was there a significant increase in rectal temperature or clinical signs observed in any of the sows. However, clinical signs indicative of CSFV infection were observed in one sow from group 5 treated with the control strain Alfort-Ems H297K. The sow experienced an abortion 14 days after vaccination and was euthanized.

Serum samples collected from all sows of groups 1, 2, 4, and 5 on study days -3, 7, 14 and at necropsy were analyzed by RT-qPCR to detect CSFV genome. All of the sows from groups 1, 2, and 4 showed negative results during the whole study. In contrast, viremia was detected in control group 5 on study days 7 and 14. 

During necropsy, 77.3% of the fetuses from control group 5 were found to be macerated, and 4.5% of the fetuses in this group were mummified. In the candidate-vaccinated groups, one mummified fetus each was found in group 1 (GD18-ddErnsHC-KARD) and group 2 (QZ07-ddErnsHC-KARD). In group 4, inoculated with QZ07-sdErnsH-KARD, two mummified and two macerated fetuses were found. An overview of the fetal appearance evaluation outcomes for groups 1–5 is summarized in Table 3. To the five sows in the negative control group on the farm of origin, 79 piglets were born. The percentage of live-born piglets for each sow ranged from 89% to 94%, except for one sow that experienced difficulties during farrowing, resulting in prolonged birth and the need for obstetric intervention. We still included this sow for reproductive performance evaluation as shown in Table 3. When the rate of normal piglets of each group was compared to the negative control group, all three valid candidate groups demonstrated equivalent reproductive performance, whereas control group 5 showed severe symptoms, evidenced by the appearance of the fetuses and the occurrence of abortion.

#### 3.3.3. No CSFV Genome Detected in Fetal Samples for Two Vaccine Candidates

One of the indicators of vertical transmission is the presence of CSF vaccine virus in susceptible fetal organs or tissues. RT-qPCR testing on tissues collected from control group 5 showed an average 99.14% positive rate across all tissues collected from all fetuses (Table 4). The concentration of viral RNA present in tissues from this group was high with a mean Ct value of 22.7. Based on clinical and viral RNA testing data, CSFV Alfort-Erns H297K as control clearly crossed the placenta and infected the fetuses, as expected.

As the next step, RT-qPCR tests were performed on tissues collected from abnormal fetuses during the necropsy of sows in groups 1, 2, and 4 as described in Section 3.3.2, revealing one doubtful result in group 2 (QZ07-ddErnHC-KARD) and negative results in all other abnormal fetuses. Therefore, those abnormal fetuses observed in vaccinated groups were likely not related to CSFV infection. 

In addition to tissues from abnormal fetuses, spleens collected from all normal fetuses were tested. One spleen sample yielded a positive result, and two spleen samples were deemed doubtful in group 2 (QZ07-ddErnsHC-KARD), whereas no positive results were observed in group 1; three doubtful results were observed in group 4. Further conventional RT-PCR testing of positive or doubtful spleen samples confirmed the presence of the E2 sequence fragment only in the positive spleen from group 2 but not in any of the doubtful samples. Subsequent testing of the other collected tissues and organs of fetuses identified as positive or doubtful in spleen samples revealed additional positive tissue samples in group 2, which further support the risk for a vertical transmission of the QZ07-ddErnsHC-KARD candidate. Therefore, the remaining tissues and organs in group 2 were not further tested. In contrast, no positive samples were detected in any remaining tissues and organs of the other two groups. Only very few doubtful results were obtained from samples other than spleens in groups 1 and 4. The detailed information is summarized in Table 4.

In addition to screening for CSFV genome in fetuses as an indication of vertical transmission, fetal serum from umbilical blood was tested for CSFV-specific antibodies. In group 1 (GD18-ddErnsHC-KARD), 142 umbilical cord blood samples were successfully collected from the 143 fetuses, and none of them showed a positive result. A similar outcome was observed in the fetuses from group 4 (QZ07-sdErnsH-KARD). In the control group, only 14 umbilical cord blood samples could be collected; however, no antibodies against CSFV were detected in any of these samples. 

### 3.4. Candidates Showed DIVA Features in Sows after Vaccination

Sow sera collected at necropsy were tested using the 6B8 DIVA dcELISA. As shown in Figure 5B, in the left two columns, sows from vaccinated groups 1, 2, and 4 showed similar positive blocking rates in the commercial IDEXX CSFV antibody test but produced no positive results in the in-house 6B8 dcELISA. In contrast, the control group, inoculated with the Alfort-Erns H297K virus lacking the KARD DIVA maker, produced positive results by both ELISAs (Figure 5B, right two columns). These findings indicate that the candidates induced only very low levels of antibodies capable of competitively inhibiting the binding of the 6B8 mAb to its epitope. Together with our previous results in piglets [34], it was thus confirmed that the 6B8 dcELISA can differentiate infected animals from animals vaccinated with the vaccine candidates.

## 4. Discussion

Numerous approaches have been employed to develop the next generation of CSF MLVs with DIVA capability [7,19]. We previously reported a novel CSF marker MLV candidate demonstrating DIVA feasibility and good safety and efficacy profiles in 3-week-old piglets [34]. In the current study, we describe three additional marker vaccine candidates based on the targeted attenuation of genotype 2.1 strains [5,6,40]. Our attenuation strategy focused on two well-characterized virulence-associated residues found in the E^rns^ and N^pro^ proteins [29,30,31]. It has been well established that mutating the RNase activity site at histidine 79 in the E^rns^ protein leads to attenuation, whereas substitution at histidine 30 results in only partial attenuation [31]. Another target for attenuation is the N^pro^ protein, which is unique to pestiviruses within the Flaviviridae family [2]. This protein is known to interfere with the host’s innate immune responses by counteracting IFN-I/III responses, enhancing the establishment of a productive infection of the host [25,26,44,45]. Studies have demonstrated that a deletion of the N^pro^ gene in CSFV results in attenuation, reducing its virulence and infectivity. Notably, such deletion does not significantly affect the replication of the virus in vitro [32]. 

Based on these research findings, we utilized three attenuation strategies on two CSFV prevalent isolates with different virulence levels to develop novel vaccine candidates. These included a double deletion of E^rns^ H79 and C171, a single deletion of E^rns^ H79, and a double deletion of N^pro^ and H79 in E^rns^. The rescue of four candidates and subsequent in vitro passaging of the rescued viruses showed efficient replication, with peak titers ranging from 10^5.1^ TCID_50_/mL to 10^7.4^ TCID_50_/mL at passage six in PK/WRL cells. The rescue and cell culture passaging were also conducted in another CSFV susceptible cell line, AI-ST, and similar stock titers were determined at the indicated peak times for the P2 viruses derived from the QZ07 strain and the P7 viruses derived from the GD18 strain. However, once the final vaccine seed is defined, it will be essential to perform comprehensive replication kinetics in the AI-ST cell line to better understand the impact of the passage level on replication characteristics and genetic stability. At the same time, the in vitro stability of the candidates was confirmed by IFA using the DIVA marker mAb 6B8 and by sequencing to verify the correctness of the introduced mutations or deletions. Up to passage 10, all recombinant viruses maintained their respective mutations with no change and showed negative staining results with the 6B8 mAb.

Any CSF vaccine candidate must be tested for safety in pregnant sows, as even strains that are found to be low virulent or avirulent in piglets have been reported to cause vertical transmission [21,46]. In the current study, the safety of these four candidates (including the previously reported candidate) was evaluated in mid-gestation pregnant sows following the guidelines in Chapter 3.9.3 “Classical swine fever” from the Terrestrial Manual of the WOAH [35]. The genetically modified Alfort-Erns H297K virus was used as a control. It is defined as a partially attenuated virus because it is avirulent to piglets but still vertically transmitted to the fetuses in pregnant sows [31]. 

The investigation of reproductive performance and vertical transmission requires the inclusion of no fewer than eight healthy sows or gilts of the same age and origin per group. Vaccination is performed between the 55th and 70th day of gestation, administering a dose not less than the maximum virus titer likely to be contained in one dose of the final vaccine. Groups with a 100% positive CSFV seroconversion rate are considered as valid. Based on these criteria, three of the tested four candidates, namely GD18-ddErnsHC-KARD, QZ07-ddErnsHC-KARD, and QZ07-sdErnsH-KARD, produced valid results. In group 3, vaccinated with GD18-ddNpro-ErnsH-KARD, only four out of seven sows showed seroconversion. This finding may indicate a lack of efficient replication in adult pigs and thus likely insufficient capacity to provide protection against CSFV infection. Although previous reports have shown that N^pro^ is not essential for CSFV replication [32], the combined deletion of N^pro^ and H79 in E^rns^ seems to have significantly impaired the replication of the virus in vivo. Obviously, the mutation of two proteins associated with innate immune evasion in CSFV appears to have resulted in low replication ability in pigs. Based on the results in this study, GD18-ddNpro-ErnsH-KARD was not considered an appropriate candidate for further development. 

No clinical signs, or fever, and normal reproductive performance in the three valid vaccine candidate groups compared to the negative control group kept on the farm of origin were observed in this study. In contrast, the control group, injected with the Alfort-Erns H297K, known to be transmitted vertically, exhibited severely impaired reproductive performance, with almost 80% of the fetuses being mummified or macerated.

In addition to assessing the safety and reproductive performance, it is crucial to be aware that transplacental transmission of CSFV may still occur even in the absence of clinical symptoms in pregnant sows [21,36,37,46]. Vertical transmission can lead to the birth of CSFV-tolerant, silently infected piglets [37,39]. These persistently infected piglets may not respond to CSF vaccination and shed the virus into the environment for extended periods of time [21,46]. Therefore, testing for the presence of CSFV in fetuses is essential to determine the safety of vaccine candidates in pregnant sows. In our study, among the three valid candidate groups, the presence of the CSFV genome tested by RT-qPCR was detected in several organs collected from group 2 (QZ07-ddErnsHC-KARD), indicating a risk for vertical transmission. In contrast, RT-qPCR yielded no positive results in any of the samples of fetuses from group 1 (GD18-ddErnsHC-KARD) or group 4 (QZ07-sdErnsH-KARD). However, six and thirteen doubtful results, respectively, were obtained in some tissues or organs in those two groups. All ambiguous reaction results had high CT values from 36.62 to 38.75, which theoretically reflect very low copy numbers of CSFV genome (a Ct of 38 equates to approximately 10 copies of viral RNA per RT-qPCR reaction). Conventional RT-PCR failed to amplify the CSFV E2 sequence from any of those doubtful samples. One possible explanation for these doubtful results could be the amplification of unspecific PCR products late in the qPCR process, leading to false positives. This assumption is supported by the fact that the six doubtful samples found in group 1 belonged to six distinct fetuses from four different litters, with no other tissue or blood samples from these fetuses or their respective mother sows testing positive for CSFV. 

In general, the GD18-ddErnsHC-KARD (group 1) and QZ07-sdErnsH-KARD (group 4) vaccine candidates demonstrated good safety profiles in pregnant sows. Consequently, further development can be focused on these two candidates. A previous report showed that a single deletion of H79 significantly attenuated the virulence of the CSFV Alfort/Tuebingen strain in piglet safety trials [31]. Therefore, it is reasonable to expect that a similar H79 deletion in the less virulent QZ07 strain would be sufficient for attenuation. 

In addition to assessing the safety of the four vaccine candidates in sows, our study also aimed to investigate their DIVA capabilities. For this purpose, the in-house DIVA 6B8 dcELISA was applied to the sow serum samples collected at the end of the study [34]. The control group infected with Alfort-Erns H297K yielded positive results in both commercial ELISA and DIVA ELISA, indicating that the DIVA ELISA can detect animals infected with wildtype virus. In contrast, the sera from the sows inoculated with the vaccine candidates were all negative in the DIVA ELISA although producing 100% positive results in the commercial CSFV E2 ELISA. Thus, it was shown that the mutation of the 6B8 epitope in the E2 gene of the vaccine candidates was sufficient to prevent the induction of antibodies against the 6B8 epitope in animals. Consequently, all vaccine candidates provide DIVA capability. This result confirms the previously reported DIVA feasibility of the GD18-ddErnsHC-KARD candidate in piglets [34].

Several aspects still require further investigation in the future. As these candidates are genetically modified live vaccine candidates, the assessment of the risk of reversion to virulence and genetic stability of the mutations in vivo is crucial, even if there is currently no indication of genetic instability or any reversion to virulence. Also, vertical transmission in pregnant sows of the final vaccine seed will have to be tested as recommended by the WOAH by showing the absence of CSFV genome in piglets after birth. Furthermore, the efficacy of the QZ07-sdErnsH-KARD candidate should be evaluated alongside the GD18-ddErnsHC-KARD candidate in a piglet model. A commercial C-strain vaccine should also be included as a benchmark in the efficacy evaluation.

## 5. Conclusions

In summary, our three targeted attenuation strategies can attenuate low or moderately virulent CSFV strains and are safe in pregnant sows. Two candidates, GD18-ddErnsHC-KARD and QZ07-sdErnsH-KARD are unlikely to cause vertical transmission. Furthermore, both candidates demonstrate potential as DIVA vaccines, as evidenced by the proprietary mAb 6B8-based dcELISA. This study, together with our previous work, constitutes a proof-of-concept for the rational design of CSF antigenically marked modified live virus vaccine candidates.

## Figures and Tables

**Figure 1 viruses-16-01043-f001:**
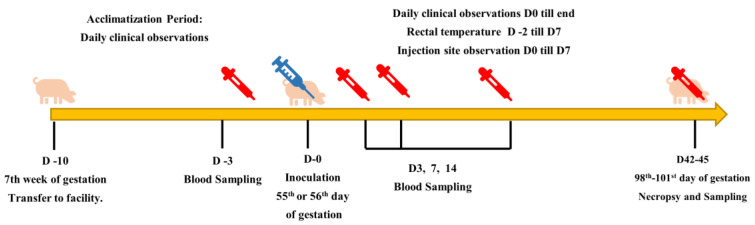
Schedule of the pregnant sow safety study.

**Figure 2 viruses-16-01043-f002:**
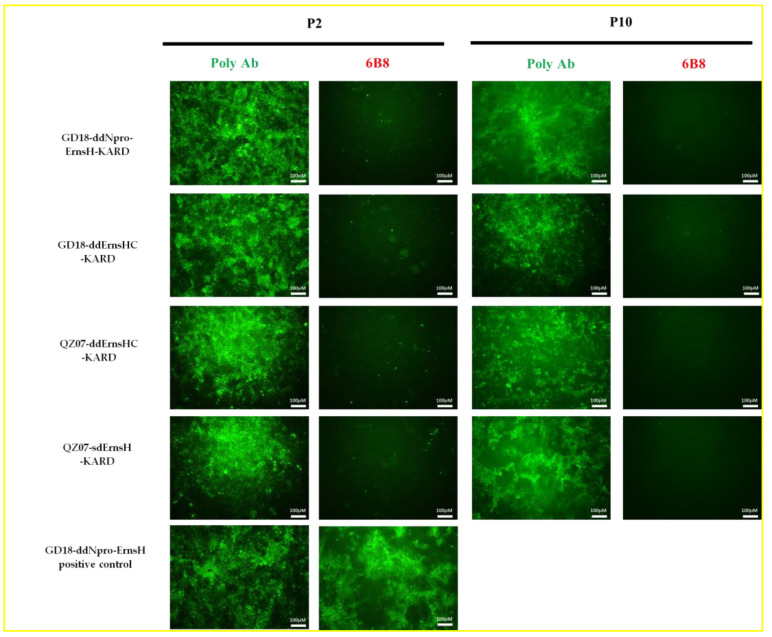
Reactivity of the four candidate viruses with the 6B8 mAb and rabbit polyclonal antibodies. Cells infected with P2 or P10 of different viruses were processed for IFA using either rabbit polyclonal antibodies against CSFV or the 6B8 mAb against glycoprotein E2. The results indicated that the KARD mutations completely abolished binding of the 6B8 mAb to cells infected with either passage of all candidates, while the polyclonal antibodies detected all candidates in the infected cells.

**Figure 3 viruses-16-01043-f003:**
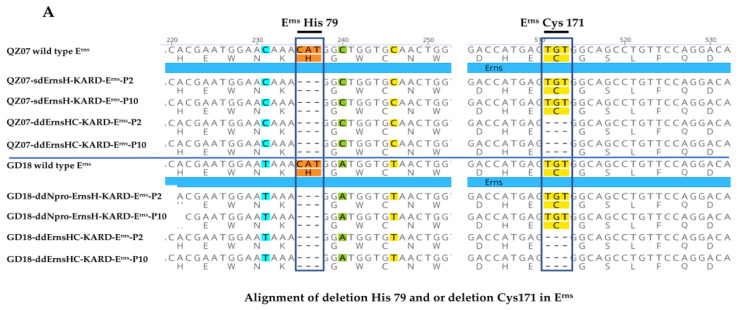
Genetic stability evaluation of the candidates during in vitro passaging. The mutations or deletions remained stable in P2 as well as P10 of all constructs. Viral genomic RNA from P2 and P10 was reverse transcribed, and the presence of the introduced mutations was confirmed by PCR and sequencing. The results were compared with the sequences of parental viruses encompassing E^rns^ at the H79 (CAT) deletion (**A**), C171 (TGT) deletion in (**A**), and the N^pro^ deletion part in (**B**). At the same time, the KARD mutations in the E2 at the S14K, G22A, E24R, and G25D mutated sites were also confirmed in (**C**).

**Figure 4 viruses-16-01043-f004:**
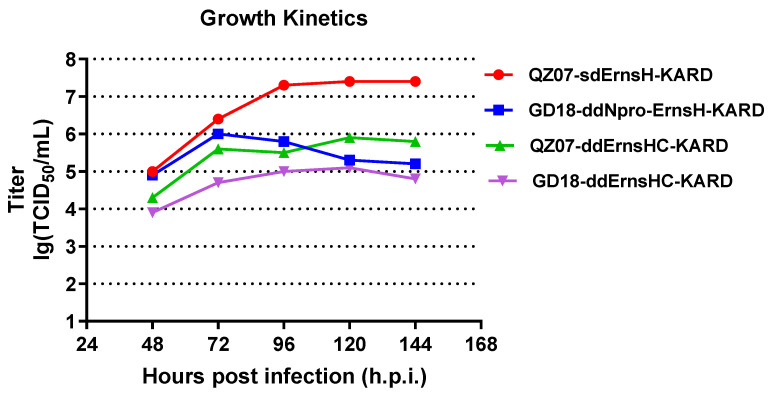
Replication kinetics in PK/WRL cells of the different vaccine candidates at passage six. PK/WRL cells were infected with each recombinant virus at an MOI of 0.01. The supernatant was collected at different hours post infection and titrated.

**Figure 5 viruses-16-01043-f005:**
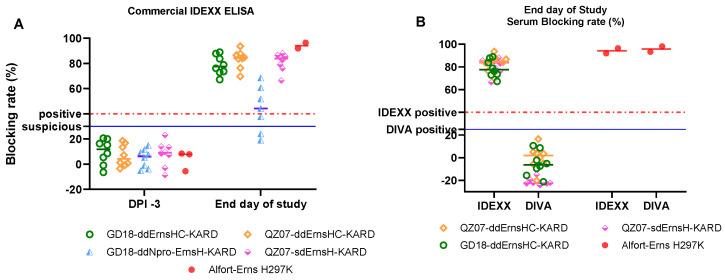
CSFV antibody responses in sows were tested by the commercial IDEXX CSFV Ab kit and the 6B8 dcELISA. The results are presented as mean antibody levels (% blocking rate). The testing results determined by the IDEXX CSFV Ab kit for sera collected at DPI -3 and the end of the study are depicted in (**A**). The positive cut-offs (dotted lines) were set at above 40% blocking rate while the suspicious cut-off values (solid line) were set at above 30% blocking rate for IDEXX ELISA. The testing results determined by the 6B8 dcELISA for sera collected at the end of the study are shown in (**B**) next to the results from the IDEXX ELISA for the same samples for direct comparison. The positive cut-off values (solid line) for the 6B8 dcELISA were set at above 25% blocking rate. Since one sow in group 5 (Alfort-Erns H297K) aborted at 14 days post vaccination and was euthanized at that day, only two serum samples were collected at the end day of study.

**Table 1 viruses-16-01043-t001:** Information on vaccine candidates constructed and investigated in this study.

Parental Strain	Virulence of Parental Strain	Vaccine Candidate	Attenuation Marker			DIVA Marker
			N^pro^	E^rns^-His (aa79)	E^rns^-Cys (aa171)	6B8 Epitope
GD18	Moderate	GD18-ddNpro-ErnsH-KARD	Deletion of amino acid (aa) residues 5–168 from N-terminus of N^pro^	Yes	No	S14 to K G22 to A E24 to R G25 to D
		GD18-ddErnsHC-KARD	No	Yes	Yes	
QZ07	Low	QZ07-sdErnsH-KARD	No	Yes	No	
		QZ07-ddErnsHC-KARD	No	Yes	Yes	

**Table 2 viruses-16-01043-t002:** Study design of the pregnant sow safety evaluation.

Group No.	Treatment Code	No. of Pregnant Sows	Vaccine Dose	Time of Treatment	Clinical Observation	Rectal Temperature	Blood Sampling	End of Study and Sampling
1	GD18-ddErnsHC-KARD	8	10^5.0^ TCID_50_, by intramuscular vaccination	55th or 56th day of gestation	Day post vaccination (DPV) 0, approx. 45	DPV -2, -1, and 0 to DPI 7	DPV -3, 3, 7, and 14 and necropsy day	DPV 42–45 (98th–101st day of gestation) Necropsy of sows for fetus condition observation Blood sampling of sows and fetuses Collection of fetus tissue samples
2	QZ07-ddErnsHC-KARD	8						
3	GD18-ddNpro-ErnsH-KARD	8						
4	QZ07-sdErnsH-KARD	8						
5	Alfort-Erns H297K	3						
6	N/A ^a^	5	N/A	N/A	N/A	N/A	Within two weeks after farrowing	Natural farrowing of newborn piglets for condition assessment

^a^. N/A stands for not applicable.

**Table 3 viruses-16-01043-t003:** Summary of reproductive performance and seroconversion rate of sows.

Group	Treatment	No. of Pregnant Sows	Total Number of Fetuses	Normal Fetuses		Mummified Fetuses		Macerated Fetuses		Sow Seroconversion by IDEXX ELISA
				No.	%	No.	%	No.	%	
1	GD18-ddErnsHC-KARD	8	143	142	99.3	1	0.7	0	0	8/8
2	QZ07-ddErnsHC-KARD	8	134	133	99.2	1	0.8	0	0	8/8
3	GD18-ddNpro-ErnsH-KARD	7 ^a^	124	124	100	0	0	0	0	4/7
4	QZ07-sdErnsH-KARD	8	128	128	96.8	2	1.7	2	1.7	8/8
5	Alfort-Erns H297K	3	44	8	18.2	2	4.5	34 ^b^	77.3	3/3
6	N/A ^c^	5	79	65	81.3 ^d^	N/A	N/A	N/A	N/A	0/5

^a^. One sow in group 3 showed severe lameness on her right front leg with obvious depression. This animal was euthanized on D14 for animal welfare reasons. Thus, the reproductive performance of group 3 was from seven sows. ^b^. Ten of the macerated fetuses were aborted by one sow in group 5 at 14 days post treatment. ^c^. N/A stands for not applicable. ^d^. The percentage of live-born piglets ranged from 89% to 94%, except for one sow, who experienced difficulties during farrowing, resulting in prolonged birth and the need for obstetric intervention. However, this sow was included for reproductive performance evaluation.

**Table 4 viruses-16-01043-t004:** Summary of RT-qPCR results of fetal tissues and blood samples.

Group	Treatment	RT-qPCR Results of Fetal Tissues and Blood Samples ^a^						Summary ^b^
		Spleen	Thymus	Kidney	Intestine	Tonsil	Umbilical Blood	
1	GD18-ddErnsHC-KARD	0/143	1D/142	1D/142	2D/142	0/143	2D/142	6D/854
2	QZ07-ddErnsHC-KARD ^c^	2D and 1P/133	3P/4	2D and 1P /4	2D and 1P/4	3P/4	1D and 2P/3	7D and 11P/152
4	QZ07-sdErnsH-KARD	3D/126	2D/128	2D/128	2D/128	2D/126	2D/124	13D/760
5	Alfort-Erns H297K	44P/44	1D and 42P/44	44P/44	43P/43	43P/43	14P/14	1D and 230P/232

^a^. The RT-qPCR was regarded as positive if a specific fluorescent signal was detected in both duplicate tests within 40 rounds of amplification (Ct < 40). A result was considered doubtful if the duplicates produced results showing one Ct value higher than 36 but lower than 40 and the other Ct value was equal to or higher than 40. P means samples identified positive in RT-qPCR and D stands for doubtful ones in RT-qPCR. ^b^. The data are shown as number of D and P samples/total tested samples. ^c^. Remaining tissues and organs in Group 2 were not further tested after positive results were identified in those three fetuses.

## Data Availability

The data presented in this study are available on request from the corresponding author.

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
