# Peer review of "Safety and DIVA Capability of Novel Live Attenuated Classical Swine Fever Marker Vaccine Candidates in Pregnant Sows"

_viruses, 2024, doi:10.3390/v16071043_

Round 1
Reviewer 1 Report
Comments and Suggestions for Authors
Abstract Line28-29
A description of the vaccine candidates “GD18-ddErnsHC- 28 KARD and QZ07-sdErnsH-KARD” is needed prior to here.
Table 1 spans across pages and is difficult to read, so it needs to be modified.
Figure 1
There is an extra character at the end of the sentence.
Materials and Methods
Line 215 Please describe more details about “Blood sample”? Whole blood or Serum? And how did you collect the samples.
Line 216, 224, 393, 495 RT-qPCR only detect small fragment of genes. Thus, “genome” should be modified to “gene”
2.5.3
Please describe more detail about the method for preparing the emulsions of the tissue samples, the RNA extraction method, and the reagents and equipment used for RT-qPCR.
2.5.4.1
Please specify the equipment used for measuring the results of ELISA.
Line 245
“infection” should be modified to “vaccination”
Line 252
“2.5.4.2.6. B8” should be modified to “2.5.4.2. 6B8”
2.5.4.2.
The criteria for interpreting ELISA results are important, so please describe them concisely.
Results
Figure 2
The following three points require correction:
1. The images are not clear, making it difficult to see the cells. Please consider modifying the images.
2. There is no scale bar.
3. There are extra characters at the end of some strings.
Figure 3
Due to extreme difficulty in readability, please consider modifying it into a simplified table. Additionally, in the caption, it is written as P2, but in the figure, it is labeled as P6, so correction is required. There is an extra figure included below Fig3A that should be removed.
Figure 4
The following three points require correction:
1. Why was the replication kinetics of P6 compared? If QZ is P2 and GD18 is P7 as actual vaccine candidates, I believe that the replication kinetics of their passage history stocks should be evaluated.
2. There are no error bars. Is this result from a single test? To ensure reliable data, it is necessary to take the average of multiple tests.
3. The strain name should be unified as "QZ07-" instead of just "QZ-" as in other parts of the text."
3.3.3.
Table 4 needs to be cited in the text.
Table 4 Line 387-388
Since the weak positive is not mentioned in either the table or the text, is there a need to define it?
Throughout the entire text, it should be consistently corrected to have superscript for Erns and Npro.
Author Response
Abstract Line28-29
A description of the vaccine candidates “GD18-ddErnsHC- 28 KARD and QZ07-sdErnsH-KARD” is needed prior to here.
Answer: We have added the information into the abstract.
Table 1 spans across pages and is difficult to read, so it needs to be modified.
Answer: We will ask the editor to modify the problems.
Figure 1
There is an extra character at the end of the sentence.
Answer: We have updated the figure and we will confirm it in the new version sent by editor.
Materials and Methods
Line 215 Please describe more details about “Blood sample”? Whole blood or Serum? And how did you collect the samples.
Answer: We have modified as whole blood sample and added the serum collection method.
Line 216, 224, 393, 495 RT-qPCR only detect small fragment of genes. Thus, “genome” should be modified to “gene.”
Answer: The CSF virus only have one full length RNA genome presented. The RT-qPCR always detects a small fragment on the full-length genome, and this is considered representative of the complete genome of the CSF virus. In viral diagnostics, detection of genome copies by qPCR or RT-qPCR is also a common practice. Thus, we think genome is a proper term to be used in this manuscript.
2.5.3 Please describe more detail about the method for preparing the emulsions of the tissue samples, the RNA extraction method, and the reagents and equipment used for RT-qPCR.
Answer: All tests were done at the World Organization for Animal Health (WOAH) and EU CSFV Reference Laboratory at the University of Veterinary Medicine Hannover, Hannover, Germany according to their validated procedure, details can be referred to the Ref. 42.
2.5.4.1 Please specify the equipment used for measuring the results of ELISA.
Answer: The ELISA testing were also conducted by the same CSFV Reference ab and the testing method were conducted strictly according to the manufacturer`s instructions and the results were measured under OD450 using a standard ELISA reader. We don’t have the specific brand or type of the equipment of the reference lab.
Line 245 “infection” should be modified to “vaccination.”
Answer: We have modified this word.
Line 252 “2.5.4.2.6. B8” should be modified to “2.5.4.2. 6B8”
Answer: We have modified this sentence.
2.5.4.2. The criteria for interpreting ELISA results are important, so please describe them concisely.
Answer: We have added the criteria and include the reference for the method.
Results
Figure 2 The following three points require correction:
- The images are not clear, making it difficult to see the cells. Please consider modifying the images.
Answer: We have modified the image to improve quality of the fluorescence micrographs.
- There is no scale bar.
Answer: The indication bar of 100μm was also added in each image.
- There are extra characters at the end of some strings.
Answer: We think this may be due to the format transfer problem, we will work with the editor to get this corrected before publication.
Figure 3
Due to extreme difficulty in readability, please consider modifying it into a simplified table. Additionally, in the caption, it is written as P2, but in the figure, it is labeled as P6, so correction is required.
Answer: Error was corrected. We will work with the editor to improve readability.
There is an extra figure included below Fig3A that should be removed.
Answer: We have modified the images and remove the extra figure.
Figure 4
The following three points require correction:
- Why was the replication kinetics of P6 compared? If QZ is P2 and GD18 is P7 as actual vaccine candidates, I believe that the replication kinetics of their passage history stocks should be evaluated.
Answer: As described in Materials and Methods, in vitro characterization and replication kinetics were performed using the same passage level P6 virus in PK/WRL cells. For the animal study, virus stocks were prepared in AI-ST A1 cells. Replication kinetics were not repeated in AI-ST A1 cells.
- There are no error bars. Is this result from a single test? To ensure reliable data, it is necessary to take the average of multiple tests.
Answer: Yes, it is the result of one test only.
For one of the candidates, GD18-ddErnsHC-KARD, it has been well characterized for in vitro growth as shown in reference 34, the passage P9 showed similar growth curve compared with P6 stock in this study. Therefore, we think our growth curve assay is repeatable and reliable. Of course, in future study, we will carefully characterize the growth kinetics with multiple test to present more reliable data.
- The strain name should be unified as "QZ07-" instead of just "QZ-" as in other parts of the text."
Answer: We have modified this word in the figure.
3.3.3.Table 4 needs to be cited in the text.
Answer: The table 4 is described and cited in the section 3.3.3, last sentence of the third paragraph.
Table 4 Line 387-388
Since the weak positive is not mentioned in either the table or the text, is there a need to define it?
Answer: Thanks for the suggestion. We also realize the “weak positive” is confusing since it might come from false positive. Therefore, we have changed it to “inconclusive” and described in part 2.5.3 as “A result was considered doubtful if the duplicates produced one inconclusive result (a Ct value between 36 and 40) and one negative result.”
Throughout the entire text, it should be consistently corrected to have superscript for Erns and Npro.
Answer: Erns and Npro were formatted as requested except for the names of the recombinant viruses.

Reviewer 2 Report
Comments and Suggestions for Authors
This manuscript reports the evaluation of immunogenicity and safety in pregnant sows of four genetically-engineered live-attenuated DIVA vaccines against classical swine fever (CSFV). One of these four vaccines was reported earlier to be efficacious, safe and allow DIVA in piglets (reference 29, Tong 2022, doi.org/10.1016/j.vaccine.2022.10.035). The authors used two different CSFV backbones (moderate and low virulent) with three different targeted attenuation strategies consisting of (i) a single deletion of H79 of Erns abolishing RNase activity, (ii) a double deletion of H79 and C171 in Erns (reported in reference 29), and (iii) a deletion of Erns H79 combined with the deletion of Npro. In addition, these constructs contain a mutated epitope in E2 allowing DIVA, as published earlier by the same group (reference 29).
General evaluation: This is an interesting follow-up study of the previous report mentioned above. The data confirm previous results, and demonstrate safety in pregnant sows of at least two variants of the live-attenuated marker vaccine published previously. The experiments are performed correctly, according the WOAH standards, except that the fetuses were analyzed before birth. A valuable aspect is the inclusion a the Alfort-Erns H297K control for transplacental infection and damage to the fetuses, which validates the data with the candidate vaccines proposed here. Nevertheless, since weak CSFV PCR signals were found in a few fetuses, a small doubt remains on whether any persistently-infected offspring can be completely excluded. Although unlikely, this requires eventually confirmation in piglets, after farrowing with a representative number of vaccinated sows. This should at least be discussed briefly as an outlook (see comment 11 below). Overall, the results are important and of general interest for CSFV vaccine development, and are to my opinion suitable for publications after minor revision.
Specific points:
Comment 1: in the abstract, lines 28-29, it would be more informative to describe the features of the vaccines and provide the vaccine names in brackets.
Comment 2: the introduction does not provide sufficient background on the different CSFV DIVA vaccines that have been reported in the literature. The review of Coronado et al, 2021 (doi.org/10.3390/vaccines9020154) is cited, but not in the context of DIVA vaccines. Although other reviews on DIVA vaccines are cited, the authors should elaborate a bit more specifically on the different CSFV DIVA vaccines in the section of lines 50 to 54. The Suvaxyn CSF Marker vaccine licensed to Zoetis should be mentioned in the introduction. One suggestion is to move the first paragraph of the discussion to the introduction.
Comment 3: on line 93, the reference to the mutated 4-amino-acid epitope for DIVA purposes is wrong. It is obviously reference 29, not 23. I did not verify all refs. The others I picked were correct. Please check/correct systematically.
Comment 4: As the authors mention, the WOAH guidelines specify that the vaccine dose should not be less than the maximum virus titer likely to be contained in one dose. What were the criteria for setting the maximum vaccine dose to 10E5 TCID50/ml? The authors should specify this in the text and mention in the discussion how this vaccine dose compares with the dose recommended for the C-strain and the Suvaxyn CSF Marker vaccine for instance.
Comment 5: Figure 2 shows reactivity of the mutant vaccines to polyclonal anti-CSFV serum that detects epitopes in Erns, E2 and NS3, as opposed to the lack of reactivity against the 6B8 mAb. Did the authors test E2-specific sera or mAbs that would detect E2 as opposed to the 6B8 mAb.
Comment 6: on lines 346-347 the authors may disclose here already that these abnormal fetuses were negative (one doubtful) for CSFV and refer to 3.3.3
Comment 7: “there is not positive samples” (line 381) is grammatically wrong.
Comment 8: on line 404, correct “CSFVV”
Comment 9: on line 421, correct “one sow at in group 5”. This sentence (lines 421-423) has other grammatical errors.
Comment 10: The number of doubtful or positive RT-qPCR samples found in the organs of the fetuses of the different groups is tendentially the opposite of what would be expected. There were more doubtful and weakly positive samples in the fetuses of group 2 compared to the fetuses of group 1. This is a bit surprising, since the only difference of the vaccines of these two groups is the backbone. According to Table 1, the backbone of the group 2 vaccine, QZ07 is low virulent, while the backbone of the group 1 vaccine, GD18 is moderately virulent. Accordingly, the opposite may be expected, i.e. slightly better replication of the GD18-derived compared to the QZ07-derived viruses with the identical mutations (ddErnsHC-KARD). Along the same line, the vaccines of group 1 and 4 differ by the mutations – double versus single deletion in Erns, respectively – and the backbone – moderate versus low virulent, respectively. Interestingly there were six versus thirteen doubtful results, in tissues of group 1 versus 4, respectively, which is again the opposite of what may be expected. One may question whether there is indeed a statistically significant difference, or whether the observed differences are simply by chance. This may be discussed in paragraph of lines 488-508.
Comment 11: related to the previous and to the general comment, the evaluation deviated from WOAH recommendations in that fetuses were tested for the presence of vaccine virus two weeks before birth, instead of post-partum testing of piglets. Since a few weak CSFV PCR signals were found in a few fetuses, especially in group 2, absence of any persistent CSFV in piglets should eventually be confirmed post-partum with a representative sample size. This is certainly beyond the scope of this study, but should be discussed at least briefly as an outlook.
Comments on the Quality of English Language
There are a few grammatical errors and typos (some but not all are mentioned in the specific comments)
Author Response
Specific points:
Comment 1: in the abstract, lines 28-29, it would be more informative to describe the features of the vaccines and provide the vaccine names in brackets.
Answer: We have added the information into the abstract.
Comment 2: the introduction does not provide sufficient background on the different CSFV DIVA vaccines that have been reported in the literature. The review of Coronado et al, 2021 (doi.org/10.3390/vaccines9020154) is cited, but not in the context of DIVA vaccines. Although other reviews on DIVA vaccines are cited, the authors should elaborate a bit more specifically on the different CSFV DIVA vaccines in the section of lines 50 to 54. The Suvaxyn CSF Marker vaccine licensed to Zoetis should be mentioned in the introduction. One suggestion is to move the first paragraph of the discussion to the introduction.
Answer: Good suggestion. We have modified both the introduction and discussion part.
Comment 3: on line 93, the reference to the mutated 4-amino-acid epitope for DIVA purposes is wrong. It is obviously reference 29, not 23. I did not verify all refs. The others I picked were correct. Please check/correct systematically.
Answer: All references have been checked and no. has been updated accordingly.
Comment 4: As the authors mention, the WOAH guidelines specify that the vaccine dose should not be less than the maximum virus titer likely to be contained in one dose. What were the criteria for setting the maximum vaccine dose to 10E5 TCID50/ml? The authors should specify this in the text and mention in the discussion how this vaccine dose compares with the dose recommended for the C-strain and the Suvaxyn CSF Marker vaccine for instance.
Answer: The Suvaxyn CSF Marker is based on a BVDV strain which would show different dose format compared with CSFV since the cell line for producing is also different. The immunization dose in our study refers to the experience in C-strain vaccine development. The commercial CSF C-strain vaccine of Boehringer Ingelheim, which launched in China presented 104.0 TCID50/dose. Our previous report demonstrated one of the candidates is efficacious at the dose of 103.0 TCID50 as shown in reference 34. Thus, the 105.0 TCID50 of candidate viruses were sufficient in safety evaluation to fulfill the 10-fold overdose requirement.
Comment 5: Figure 2 shows reactivity of the mutant vaccines to polyclonal anti-CSFV serum that detects epitopes in Erns, E2 and NS3, as opposed to the lack of reactivity against the 6B8 mAb. Did the authors test E2-specific sera or mAbs that would detect E2 as opposed to the 6B8 mAb.
Answer: We have the data for a reference CSFV E2 mAb targeting antigenic domain A/D, and it showed the same reaction pattern as polyclonal anti-CSFV serum did.
Comment 6: on lines 346-347 the authors may disclose here already that these abnormal fetuses were negative (one doubtful) for CSFV and refer to 3.3.3
Answer: We would like to put all RT-qPCR results together and used 3.3.2 paragraph to interpret vertical transmission data package.
Comment 7: “there is not positive samples” (line 381) is grammatically wrong.
Answer: We have modified this sentence as no positive samples.
Comment 8: on line 404, correct “CSFVV”
Answer: We have modified this as CSFV.
Comment 9: on line 421, correct “one sow at in group 5”. This sentence (lines 421-423) has other grammatical errors.
Answer: We have modified this sentence as one sow in group 5.
Comment 10: The number of doubtful or positive RT-qPCR samples found in the organs of the fetuses of the different groups is tendentially the opposite of what would be expected. There were more doubtful and weakly positive samples in the fetuses of group 2 compared to the fetuses of group 1. This is a bit surprising, since the only difference of the vaccines of these two groups is the backbone. According to Table 1, the backbone of the group 2 vaccine, QZ07 is low virulent, while the backbone of the group 1 vaccine, GD18 is moderately virulent. Accordingly, the opposite may be expected, i.e., slightly better replication of the GD18-derived compared to the QZ07-derived viruses with the identical mutations (ddErnsHC-KARD). Along the same line, the vaccines of group 1 and 4 differ by the mutations – double versus single deletion in Erns, respectively – and the backbone – moderate versus low virulent, respectively. Interestingly there were six versus thirteen doubtful results, in tissues of group 1 versus 4, respectively, which is again the opposite of what may be expected. One may question whether there is indeed a statistically significant difference, or whether the observed differences are simply by chance. This may be discussed in paragraph of lines 488-508.
Answer: In an ideal situation no doubtful results would be found at all. However, every testing method has a chance of false negative and false positive, as well as a limit to what it can detect. That's why 8 pregnant sows in each group were included and as many organs of fetuses as possible were tested for the presence of the CSFV genome in this study. The testing method was carefully set up by the CSFV reference lab, and the method is well validated. For any samples that gave doubtful results, a second round of RT-PCR for another gene in CSFV was done, and no positive results were obtained. We thus think that the doubtful results might come from nonspecific amplification as it is discussed. On the basis of the current doubtful results, a definite conclusion on the best candidate cannot be drawn. However, for the group QZ07-ddErnsHC-KARD, we did amplify and confirm the CSFV sequence by conventional RT-PCR and sequencing, so we can conclude that this candidate caused vertical transmission. This study served as the basis and direction to better understand the safety profile of those candidates in the most stringent and sensitive pregnant sow model for future development. We might continue the research to understand why double deletion showed a high potential for vertical transmission, but that is not the focus of this study.
Comment 11: related to the previous and to the general comment, the evaluation deviated from WOAH recommendations in that fetuses were tested for the presence of vaccine virus two weeks before birth, instead of post-partum testing of piglets. Since a few weak CSFV PCR signals were found in a few fetuses, especially in group 2, absence of any persistent CSFV in piglets should eventually be confirmed post-partum with a representative sample size. This is certainly beyond the scope of this study but should be discussed at least briefly as an outlook.
Answer: Thanks for your good suggestion. In this study we did not want to have sows give birth to potentially infected piglets and therefore decided to deviate from the WOAH recommendations to avoid suffering in the piglets. We will definitely confirm this key outcome when final vaccine seed is defined. We also add one sentence of description as an outlook in the conclusion paragraph.
Comments on the Quality of English Language:There are a few grammatical errors and typos (some but not all are mentioned in the specific comments)
We have corrected all typos and corrected all the grammatical errors by cross check by all authors.

Round 2
Reviewer 1 Report
Comments and Suggestions for Authors
The replication kinetics and pathogenicity of the virus can change depending on the passage history of the virus stock. Since the significance of evaluating the proliferation at P6, rather than the actual vaccine strain's passage history, is unclear, this should be explained. Additionally, to clarify the reliability of the results, it should be noted that the data is based on a single test.
Author Response
Answer: It is a good suggestion. Some explanations were added into lines 356-361 in the result and in the discussion from lines 508-514. At this moment, it is at the stage of leading candidates screening. Once the final vaccine seed is defined, it will be essential to perform comprehensive replication kinetics and process development to better understand the impact of the passage level on replication characteristics and genetic stability. It is also noted that the growth kinetics data is based on a single test in lines 170 and 346 in both methods and results sections.